# Comorbidity of age-related macular degeneration with Alzheimer's disease: A histopathologic case-control study

Gordon J. Smilnak[1]☯, John R. Deans[1]☯¤, P. Murali Doraiswamy[2], Sandra Stinnett[1], Heather E. Whitson[3], Eleonora M. Lad[1]*

**1** Department of Ophthalmology, Duke University Medical Center, Durham, NC, United States of America, **2** Department of Psychiatry and Behavioral Sciences, Duke University Medical Center, Durham, NC, United States of America, **3** Department of Internal Medicine, Division of Geriatrics, Duke University Medical Center, Durham, NC, United States of America

☯ These authors contributed equally to this work.
¤ Current address: Department of Ophthalmology, Kittner Eye Center, University of North Carolina at Chapel Hill, Chapel Hill, NC, United States of America
* nora.lad@duke.edu

**Data Availability Statement:** All relevant data are within the paper.

**Funding:** This work was supported by the National Institutes of Health grant number K23EY026988

## Abstract

### Introduction

Previous studies evaluating the association between clinically diagnosed Alzheimer's disease (AD) and age-related macular degeneration (AMD) have generated conflicting results. This study is the first to assess whether AMD prevalence is higher in AD patients than non-AD controls by using histopathology to definitively diagnose AD.

### Methods

This was a retrospective case-control study utilizing diagnostic information extracted from autopsy reports of patients age 75 and above, including 115 with a neuropathological diagnosis of AD and 57 age-matched normal controls.

### Results

The rate of AMD was not significantly higher in AD cases (53.0%) than in controls (59.6%) (z = 0.820, p = 0.794). AMD severity as determined by Sarks score was similar between AD patients and controls ($\chi^2 = 2.96$, p = 0.706). There was also no significant association between Braak stage of AD severity and AMD ($\chi^2 = 4.55$, p = 0.602).

### Discussion

No significant effect of AD diagnosis or pathologic severity on AMD comorbidity was found, suggesting that any shared mechanisms between AMD and AD may be nondeterministic.

(EML). The funders had no role in study design, data collection and analysis, decision to publish, or preparation of the manuscript.

**Competing interests:** The authors have declared that no competing interests exist.

## Introduction

Alzheimer's disease (AD) and age-related macular degeneration (AMD) are common neuro-degenerative diseases associated with advanced age which share numerous pathological and mechanistic features [1–3]. Notably, both AD and AMD are histopathologically characterized by abnormal extracellular deposits. In AD, senile plaques composed primarily of amyloid-β (Aβ) form throughout the central nervous system (CNS) cortex and hippocampus [2]. These plaques are associated with neuronal dysfunction and cell death leading to progressive cognitive decline and memory loss. In AMD, lipoproteinaceous immune deposits called drusen form between the retinal pigment epithelium (RPE) and Bruch's membrane (BrM). The presence of drusen is associated with photoreceptor dysfunction and loss through either an atrophic ("dry" AMD) or exudative ("wet" AMD) process that results in progressive visual decline. Interestingly, senile plaques and drusen have been shown to have many common constituents, including Aβ, apolipoprotein E (APOE), and complement immune components [4–7]. These findings, along with shared environmental risk factors for AD and AMD including advanced age, cigarette smoking, and hyperlipidemia associated with a Western diet, have led to the hypothesis that these diseases may have a common underlying pathophysiology [1]. The presence of abnormal extracellular deposits, specifically senile plaques in AD and drusen in AMD, may result in chronic inflammation and oxidative stress that damage surrounding tissues; however, the elucidation of precise mechanisms and development of targeted treatments have remained elusive.

Demonstrating an association or lack of association between AD and AMD would provide the rationale for future work investigating common and divergent features of AD and AMD, as well as drive therapeutic translation across diseases. For example, it may suggest the addition of clinical endpoints to ongoing trials related to both conditions to determine whether therapeutic agents for one condition have any treatment effect on the other condition. It would also justify longitudinal studies aimed to better understand whether these diseases co-develop synchronously or whether one disease may represent a risk factor for the other. Furthermore, patient care would be impacted, as health care providers could inform patients of the likelihood of increased risk for associated disease given a single diagnosis of either AD or AMD.

Unfortunately, demonstrating an association between AD and AMD has proven difficult, and previous studies have generated conflicting results [8–14]. A major limitation of all prior work investigating the association of AD and AMD was the reliance on clinical criteria to diagnose AD. There are standardized clinical criteria for AD diagnosis (National Institute of Aging-Alzheimer Association Criteria), and research on imaging, blood, and cerebrospinal fluid (CSF)-based biomarkers to support the clinical impression is rapidly advancing [15]. However, the gold standard for AD diagnosis remains the neurohistopathology analysis. A recent study showed that the sensitivity and specificity of a clinical diagnosis of "probable AD", representing the highest level of confidence under the previously-used National Institute of Neurological and Communicative Disorders and Stroke (NINCDS) criteria, is approximately 71% when compared to the gold standard of neuropathological diagnosis [16]. Furthermore, the AD population in previous studies has potentially been biased towards milder disease, as patients with advanced dementia may be less likely to seek or receive ophthalmic or other healthcare [11, 17, 18].

Given these limitations of previous studies, the purpose of our work was to employ histopathological analysis of eye and brain autopsy specimens with the main goal to determine whether the comorbidity rate of AMD is significantly greater in patients with a neuropathological diagnosis of AD than in age-matched patients without AD. The main advantage of this

methodology is that it employs the gold-standard pathological diagnosis for AD and evaluation of stages of AMD and AD. A second advantage is the ability to sample the full spectrum of AD. In addition to the primary goal, analysis of eye and brain pathology reports along with patient records enabled investigation of the association between AD and other degenerative diseases including glaucoma and other dementias.

## Methods

### Histopathology specimens

Pathologic specimens of eyes and brains of autopsy subjects aged 75 and above that presented to Duke University Medical Center were prepared and histologically analyzed by methods previously described [19, 20]. This age cutoff was chosen because the incidence of both AMD and AD increase significantly after age 75 [21, 22]. AMD severity was graded in a minimum of 5 eye sections per eye by a board-certified ophthalmic pathologist (AL) as previously detailed by Sarks [23] [Fig 1]. AD was graded in brain specimens by neuropathologists as previously described by Braak and Braak [24] and in accordance with the National Institute on Aging-Alzheimer's Association and the Consortium to Establish a Registry for Alzheimer's Disease (CERAD) guidelines [12, 25]. For the purposes of our study, AMD was defined as Sarks grades III-VI, corresponding to intermediate to severe clinical AMD; the diagnosis of AMD was further classified as either "early" (Sarks grades III & IV) or "late" (Sarks grades V & VI). For cases with discrepant stages between the two eyes, the higher Sarks score for AMD was considered in the analysis. AD was defined as Braak and Braak Stage III-VI with "moderate" or "frequent" neuritic plaque density, representing CERAD 2 or 3 scores [12]. AD cases were further grouped into the categories of "Early AD" (Braak and Braak Stages III-IV) and "Late AD" (Braak and Braak Stages V-VI). The histopathologic diagnosis of advanced glaucoma was made when the following were observed: sparse retinal ganglion cells, diminished size of optic nerve axon bundles, and fibrotic thickening or "cupping" of the optic nerve [26]. "Non-AD dementia" included any other dementia with neuropathology features characteristic of fronto-temporal degeneration, Lewy body disease, Parkinson's disease, or hippocampal sclerosis. Cerebrovascular atherosclerosis (CVA) was defined as the evidence of moderate to severe CNS atherosclerosis. The use of autopsy eyes and brain tissue for research was approved by the Institutional Review Board of Duke University. The need for participant consent was waived by the Institutional Review Board for this decedent research, as all research subjects are

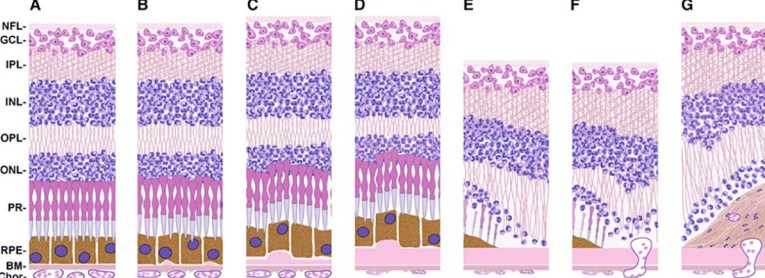

**Fig 1. Diagram illustrating the Sarks stages of AMD. a** Sarks I: normal control. **b** Sarks II: few small drusen. **c** Sarks III: early AMD with thin continuous sub-RPE deposits. **d** Sarks IV: intermediate AMD with thick sub-RPE deposits overlying degenerating choriocapillaries. **e** Sarks V: geographic atrophy. **f** Sarks V: choroidal neovascularization. **g** Sarks VI: disciform scar. NFL = Nerve Fiber Layer, GCL = Ganglion Cell Layer, IPL = Inner Plexiform Layer, INL = Inner Nuclear Layer, OPL = Outer Plexiform Layer, ONL = Outer Nuclear Layer, PR = Photoreceptors, RPE = Retinal Pigment Epithelium, BM = Bruch's Membrane, chor = choroid. Reprinted under a CC BY license, with permission from publisher SpringerNature, original copyright 2014 [27].

deceased, and all personal health information was used solely for research and was not disclosed to anyone outside Duke University without removing all identifiers.

### Chart review

Patient data including demographic factors and comorbidities were obtained from the patient chart in the EPIC MaestroCare electronic medical record system of Duke University Medical Center. Patient comorbidities (cardiovascular disease, hypertension, diabetes, depression) were defined by their presence in the patient's medical problem list.

### Statistical analyses

An independent samples t-test was performed to compare the average age of patients in the AD and control study groups. A one-tailed z-test for two population proportions was conducted to determine any significant difference in AMD co-prevalence between the AD and control cohorts; two-tailed tests were used to assess all other demographic factors and potential disease associations. In addition, chi-squared tests of independence were performed to assess the comorbidity rate of AMD and AD, as well as the relationship between severity of each disease in the presence of the other. To further characterize the likelihood of AMD given an AD diagnosis, cases were divided into "Early AD" (Braak and Braak Stages III-IV) and "Late AD" (Braak and Braak Stages V-VI) cohorts, and odds ratios with 95% confidence intervals were calculated between each of these groups. Significance was defined as $p < 0.05$ for all analyses. The primary analysis was the comparison between AMD prevalence in AD vs. controls. As other analyses were considered exploratory, a correction for multiple comparisons was not applied in the secondary analyses. Analyses were performed in SAS 9.3 (SAS Institute, Cary NC).

## Results

Eye and brain samples from 115 AD and 57 control patients were identified and histopathologically staged for AMD and AD. Among the control cohort, 36 cases were Braak & Braak Stage 0, 7 were Stage I, and 14 were Stage II; within the AD cohort, 42 cases were Braak Stage III, 30 were Braak Stage IV, 31 were Braak Stage V, and 12 were Braak Stage VI [Table 1]. Overall the age and race distributions between the two groups were similar (t = -0.211, p = 0.833; AD mean 86.8 years, ages 75–103, 86.1% white vs. control mean 86.6 years, ages 76–101, 78.9% white) [Table 1]. The gender distribution was significantly different between the groups, with a higher preponderance of female patients in the AD than the non-AD group (p = 0.032, AD 66.1% female, control 49.1% female) [Table 1]. The two cohorts were found to be similar with respect to co-morbidities including diabetes (p = 0.232) and hypertension (p = 0.353) [Table 1].

There was no significant difference in AMD rate between AD patients and controls (z = 0.820, p = 0.794; 53.0% vs. 59.6%) [Table 2]. Furthermore, AMD severity as determined by Sarks score was similar between AD patients and controls ($\chi^2 = 2.96$, p = 0.706), or between "early" and "late" AD ($\chi^2 = 5.60$, p = 0.848) [Fig 2]. Likewise, we did not find an association between Braak staging of AD and AMD ($\chi^2 = 4.55$, p = 0.602), even when stratified by "early" and "late" AD ($\chi^2 = 5.16$, p = 0.523) [Table 3, Fig 3].

A secondary goal of this study was to employ the database of cases with matched eye and brain pathology reports to investigate the prevalence of glaucoma and other neuropathological diagnoses in the cohort of AD patients and non-AD controls over age 75 [Table 2]. Comorbidity rates of advanced glaucoma, the only stage of glaucoma that can be reliability diagnosed via histopathology analysis, were similar between AD patients and controls (13.9% vs. 17.5%,

**Table 1. Demographic characteristics of the cohort.**

| | Control (n = 57) | AD (n = 115) | p-value |
|---|---|---|---|
| **Demographic data** | | | |
| Age (years, SD) | 86.6 (5.9) | 86.8 (5.7) | 0.833 |
| Gender (% female) | 49.1 | 66.1 | 0.032 |
| Race (% white) | 78.9 | 86.1 | 0.232 |
| **Systemic chronic disease data (%)** | | | |
| Diabetes | 21.0 | 13.9 | 0.232 |
| Hypertension | 43.8 | 36.5 | 0.353 |
| Depression | 7.0 | 13.0 | 0.235 |
| Smoking | 0.0 | 1.7 | 0.316 |
| **Braak & Braak AD staging (%)** | | | |
| Stage 0 | 63.2 | - | - |
| Stage I | 12.3 | - | - |
| Stage II | 24.6 | - | - |
| Stage III | - | 36.5 | - |
| Stage IV | - | 26.1 | - |
| Stage V | - | 27.0 | - |
| Stage VI | - | 10.4 | - |

p = 0.412). Likewise, frequency of cerebrovascular atherosclerosis and other dementias (Lewy body, Parkinson's disease, frontotemporal, or hippocampal sclerosis) did not differ significantly between the groups [**Table 2**].

## Discussion

To our knowledge, this represents the first study exploring the association between AD and AMD in a population age 75 and older which employed histopathological analysis to provide a definitive postmortem diagnosis and staging of these two important aging diseases. Overall, the comorbidity rate of AMD observed in both the aged AD and the non-AD groups in this study was greater than 50%, and the average patient age at death was approximately 87 years. This was higher than previous estimates that the combined prevalence of early and late AMD in white Americans and Europeans aged 85 years and older approached 45% [1, 28]. Interestingly, our cohort was approximately 20% non-white. Though this difference can be attributed to the increased sensitivity of histopathology over that of the clinical exam in diagnosing AMD, it also raises the question of whether AMD is under-diagnosed in non-white populations.

Notably, there was no significant difference in AMD comorbidity between AD and non-AD subjects. This result supports the conclusion that, in spite of the mechanistic and

**Table 2. Ophthalmic and neuropathologic diagnoses.**

| | Control (n = 57) | AD (n = 115) | p-value |
|---|---|---|---|
| **Ophthalmologic diagnoses (%)** | | | |
| **AMD** | **59.6** | **53.0** | **0.794** |
| Severe glaucoma | 17.5 | 13.9 | 0.412 |
| **Neuropathologic diagnoses (%)** | | | |
| Non-AD dementia | 61.4 | 53.9 | 0.351 |
| Cerebrovascular Atherosclerosis | 54.5 | 61.7 | 0.355 |

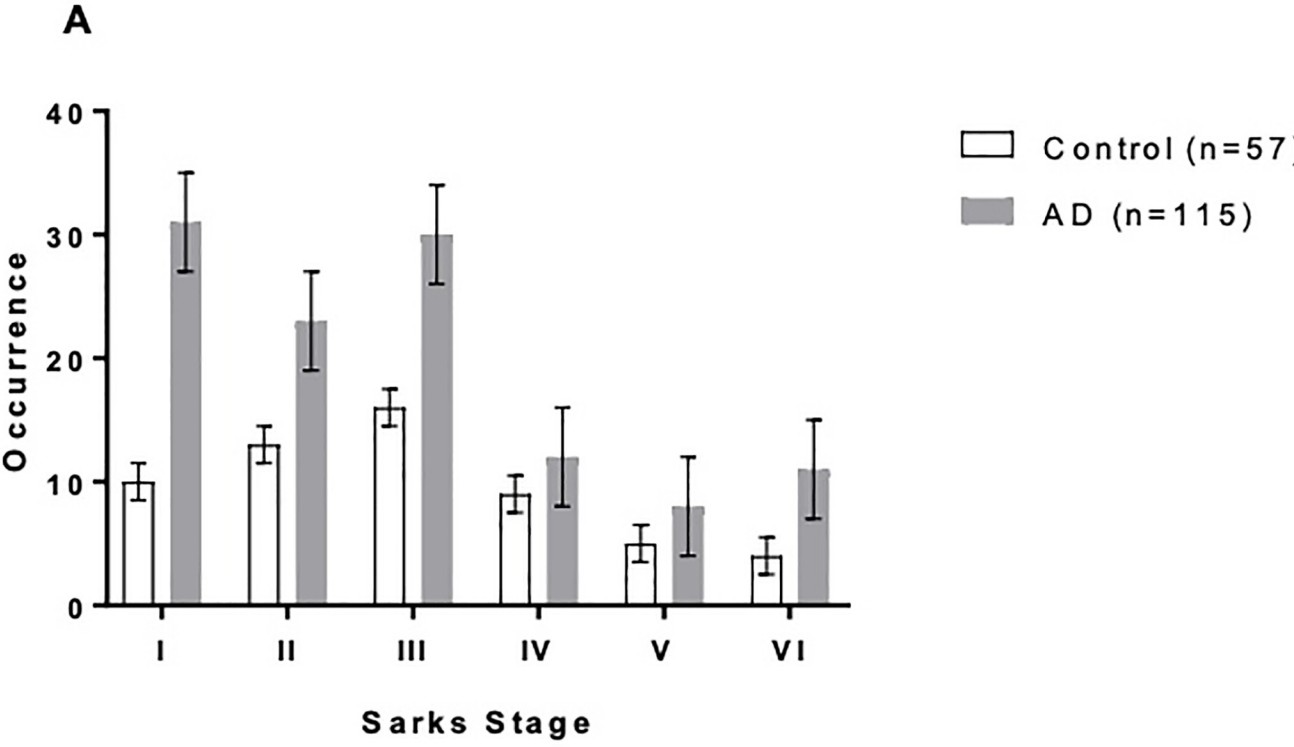

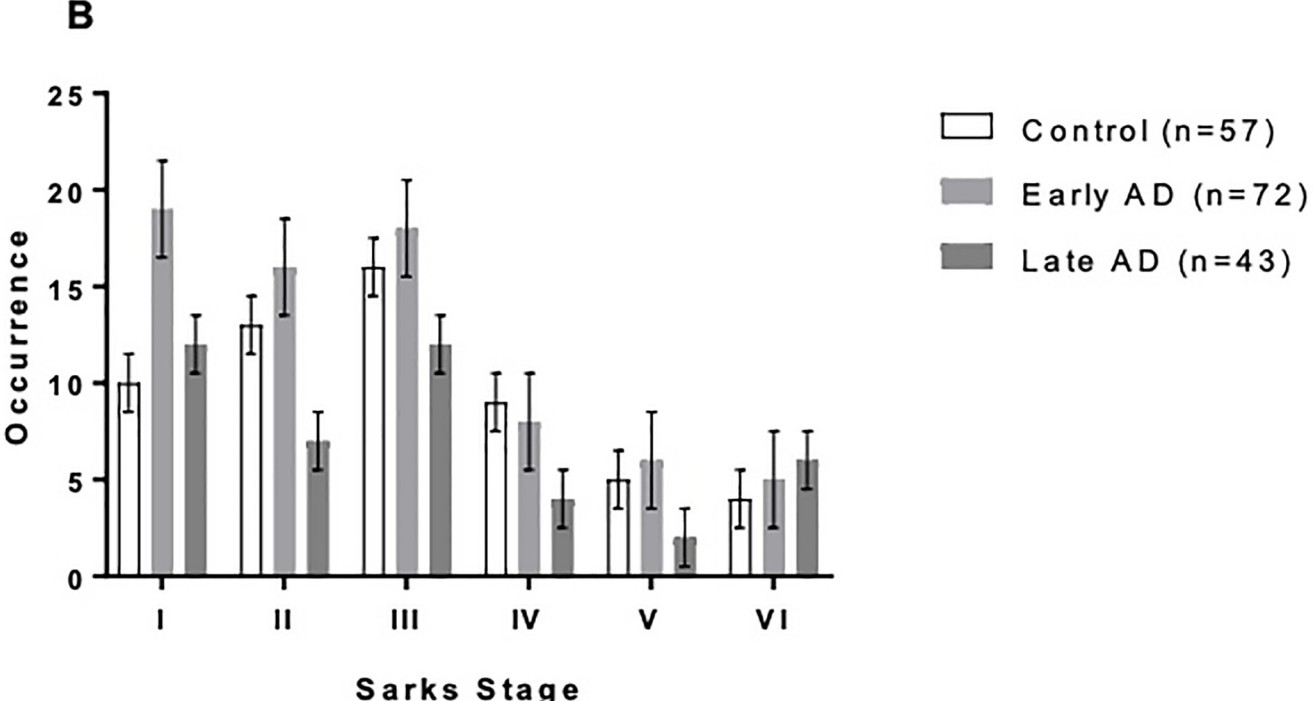

**Fig 2. AMD severity in Alzheimer's disease patients vs. controls. A** There was no significant difference in AMD severity between AD patients and controls across the spectrum of AMD as characterized by Sarks staging. AMD was defined as Sarks stages III-VI. A chi-squared test of independence was used to determine whether there was a correlation between AD diagnosis and Sarks score and was not significant at p = 0.05. **B** No significant relationship between AMD severity and AD was observed even after classifying AD cases into "early" (Braak and Braak stages III & IV) and "late" AD (Braak and Braak stages V & VI).

histopathological similarities between AD and AMD, these diseases are not associated with each other at the individual level. Our results are in agreement with two recent studies that employed markedly different methodologies. In a large record linkage study of hospital admissions, Keenan et al. included 65,894 patients in the AMD cohort, 168,092 patients in the AD cohort and used a reference cohort of over >7.7 million patients, all constructed from English National Health Service electronic records [11]. Most AMD hospital admissions in this study consisted of patients with neovascular AMD receiving intravitreal anti-VEGF therapy. This study did not find an increased risk of a subsequent hospital admission for AMD given a diagnosis of AD. In fact, there was a reduced risk of admission attributed to possible barriers to care for patients with dementia. A limitation of this study was that it mainly examined the potential association between AD and the neovascular form of AMD and not between the most common dry form of AMD. The authors further acknowledge that this AD cohort was likely to include other types of dementia, given the possible variability in diagnostic coding by various physicians. Strengths of this work included its large sample size and controlling for potential confounding factors such as socioeconomic status.

Similarly, in a large case-control study, Williams and colleagues found no increased prevalence of AMD in AD cases versus controls after correcting for age alone or age, smoking, and *APOE* ε4ε4 or ε3ε4 genotype [10]. This particular work included 258 AD cases defined by NINCDS criteria and 322 control patients. AMD grade was determined through masked examination of dilated retinal photographs centered on the macula using a modification of the system employed in the Rotterdam studies [29]. Weaknesses of this study included the fact that a greater proportion of AD cases (19.4%) than controls (12.1%) had ungradable retinal photographs. Indeed, if advanced AMD was over-represented amongst the ungradable AD group, the association between AD and AMD could have been under-estimated. Furthermore, a clinical diagnosis alone is not sufficient to determine the co-prevalence between AMD and AD. Clinically diagnosing AD is difficult and requires access to specialists and numerous imaging and laboratory tests [15]. Previous work supports that some individuals exhibit extraordinary cognitive resilience in that they do not present clinically with cognitive changes despite substantial burden of pathological AD changes in the brain. Furthermore, the AD population in previous clinical studies had a high likelihood of being biased towards milder disease, as patients with advanced dementia are less likely to receive coordinated care [11, 17, 18].

The Rotterdam study by Klaver et al., which measured the rate of incident clinically diagnosed AD in 1,438 patients with known AMD status over a four-year period, revealed that patients with advanced AMD had an increased risk of incident AD (RR = 2.1, 95% CI: 1.1–4.3). However, this increase became insignificant after controlling for age, gender, smoking, and

**Table 3. Association between AD stages and AMD.**

|  | n | AMD Prevalence (%) | Odds Ratio (95% CI) | p-value |
|---|---|---|---|---|
| **No AD (Braak 0-II)** | 34 | 59.6 | 1.00 | - |
| **Early AD (Braak III-IV)** | 37 | 51.4 | 0.72 (0.35–1.44) | 0.350 |
| **Late AD (Braak V-VI)** | 24 | 55.8 | 0.85 (0.38–1.90) | 0.700 |
| **All AD (Braak III-VI)** | 61 | 53.0 | 0.76 (0.40–1.45) | 0.413 |

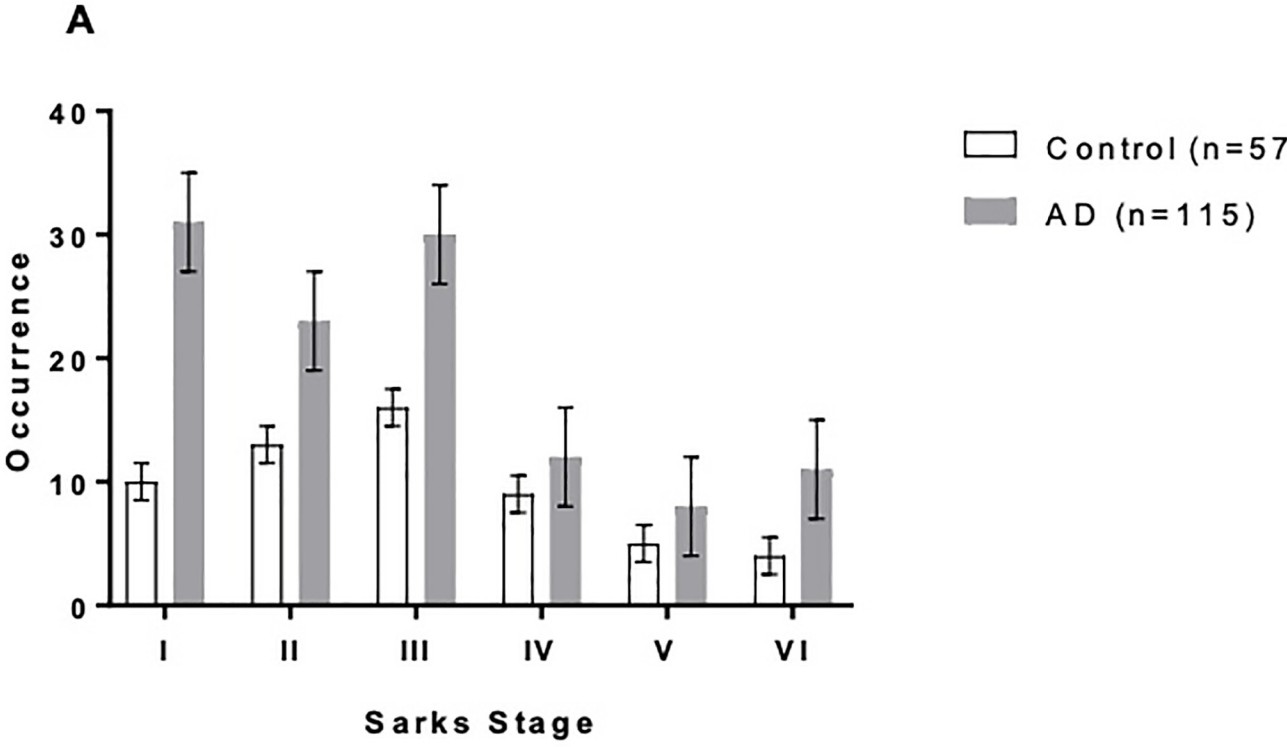

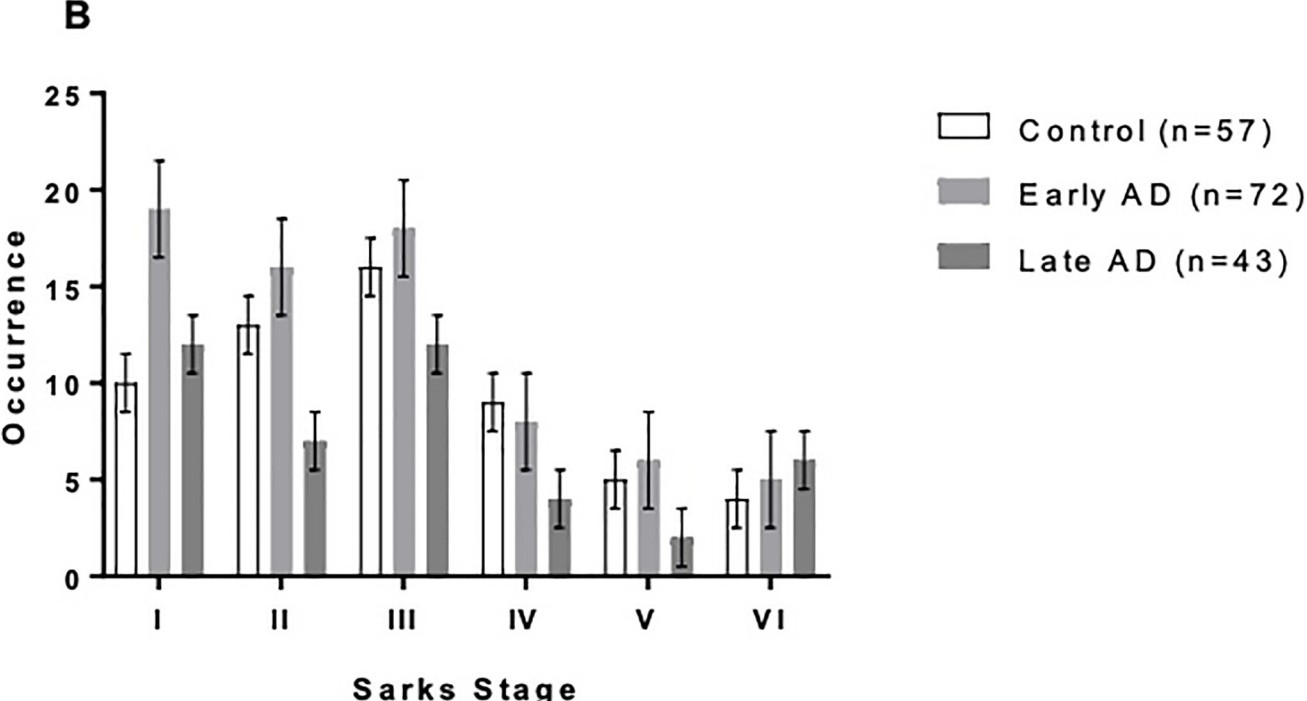

**Fig 3. AMD prevalence by Braak and Braak Alzheimer's staging. A** The prevalence of AMD did not increase with Alzheimer's severity by Braak and Braak criteria. The prevalence of AMD was highest (71.4%) in patients without AD per histopathologic criteria (Braak and Braak stages I-II) and was similar between control patients (59.6%) and severe AD patients (55.8%; Braak and Braak V-VI). No association was found between AMD prevalence and Braak and Braak AD stage by a chi-squared test of independence (p = 0.05). **B** No significant relationship between AD severity and AMD was observed even after further characterizing AMD into "early" (stages III & IV) and "late" disease (stages V & VI).

atherosclerosis (RR = 1.5, 95% CI: 0.6–3.5) [8]. A number of additional studies have suggested a possible association between AMD and cognitive impairment; however, none of these publications specifically addressed the biases of a clinical diagnosis of Alzheimer dementia [9, 13, 14].

While an association between vision loss and cognitive impairment clearly exists, our current results and the two large case control studies cited above suggest a lack of a substantive co-prevalence between AD and AMD diagnoses [10, 11, 30]. Taken together, these findings support the notion that the epidemiological link between AMD and cognitive impairment arises because vision loss affects cognitive processes or the presentation of cognitive decline, rather than that the two diseases arise concurrently due to shared underlying pathology [31]. Genetic evidence also supports the notion that AD and AMD may have distinct pathophysiology in spite of their biochemical and histological similarities. AD and AMD appear to have essentially independent genetic risk profiles [1, 32–34]. Major AMD genetic risk associations including polymorphisms in complement factor H, locus LOC387715/age-related maculopathy susceptibility 2 (*ARMS2*), and high-temperature requirement factor A1 (*HTRA1*) are not associated to AD [1, 33, 35]. Furthermore, *APOE* genotype appears to have opposite effects on disease risk profiles between AD and AMD. The *APOE*-ε4 allele has been shown to increase the risk of AD and cardiovascular disease but to be protective for AMD, while the converse appears to be true for the *APOE*-ε2 allele [1, 3, 36]. Our results, which show no significant association between AD and AMD at the individual level, are concordant with these diseases having separate genetic risk profiles.

Our results have additional potential implications for AD and AMD therapeutics. Presuming numerous common mechanistic features between these diseases, there has been optimism that therapies developed for one could be translated effectively to the other. Indeed, in a mouse model of AMD, an anti-Aβ therapy originally developed for AD prevented visual functional decline and development of AMD-like pathology [37]. Despite their success in animal models, anti-amyloid therapies have been less promising in humans, with recent high-profile failures of phase 2 and 3 trials in AMD and AD; however, some trials are still ongoing in AMD and patients at risk of AD development but without symptoms of dementia [38–40]. Our current results indicate that translation of anti-amyloid therapeutics successfully from AD to AMD or vice versa may be more challenging than previously suggested by animal studies.

In addition to the primary goal to evaluate the association between AD and AMD, we sought to fully utilize the dataset of matched eye and neuropathology pathology reports to assess potential associations of AD with other eye and CNS pathology. Similar to AMD, glaucoma is a common, multifactorial disease associated with aging that represents a leading cause of irreversible vision loss. Glaucoma is a progressive optic neuropathy that is generally considered to be a neurodegenerative disease given its characteristic features such as the selective loss of neuron populations, specifically of the retinal ganglion cells, and trans-synaptic degeneration occurring in the lateral geniculate nucleus and visual cortex [41, 42]. Some researchers have postulated that glaucoma may be associated with CNS neurodegenerative diseases and AD in particular; indeed, a significantly increased prevalence of glaucoma in AD patients compared to non-AD controls has been suggested by small clinical studies conducted in Europe by Bayer et al. (26% in AD vs. 5% in controls) and Japan by Tamura et al. (24% vs. 10%) [43, 44]. The current study did not show any significant difference in prevalence of advanced glaucoma

between AD patients and controls. Elucidating any potential association between glaucoma and AD remains an opportunity for further research.

## Strengths and limitations

The current work was limited by the sample size, which was smaller than for other clinical or health outcomes studies. This was due to the inclusion of autopsy patients over age 75, an age chosen *a priori* with the goal to study the population affected by aging diseases such as AMD and AD, as well as the access to cases for which both eyes and brain specimens with neuropathology characterization were also available. Second, we were not able to correct for certain known AD and AMD risk associations such as *APOE* genotype, as this history was unavailable for a large number of the autopsy cases analyzed. Third, this study relied on Braak and Braak staging to determine severity of AD. Though this method is commonly used in exploratory research studies, future studies may benefit from leveraging other pathologic criteria such as the Thal stages of amyloid deposition, positron emission tomography and *in vivo* amyloid imaging techniques currently in development [45, 46].

Finally, we noted a number of differences in demographics and comorbidities between the AD and control cohorts. The AD cohort comprised a significantly greater female proportion. This difference was considered acceptable, as previous work has shown that while AMD prevalence increases exponentially with age, there is no significant difference by gender [47]. The incidence of both diabetes and hypertension was significantly higher in the control than the AD group. This was unexpected, as some studies have suggested that diabetes and hypertension are risk factors for AD [48, 49]. Since the comorbidity data relied on chart review, it is possible that AD patients were less likely to have hypertension or diabetes noted in their problem list than control patients and that these comorbidities were not actually unequally represented between the groups. In addition, the evidence implicating hypertension or diabetes as risk factors for AMD remains weak and inconsistent [28, 50]. To control for the imbalances in demographics and comorbidities noted between the AD and non-AD group, a future direction would be the calculation of AMD prevalence from the electronic data in a cohort of control patients with the same demographics as the AD group.

The strengths of the current study lie in the use of an unbiased sample of patients presenting for autopsy, utilization of the gold standard histopathologic diagnosis of AD and ability to sample the full spectrum of both AD and AMD stages through pathology characterization.

In conclusion, this study represents the first comorbidity study exploring the association between AD and AMD that used an unbiased sample generated through histopathological analysis to definitively diagnose AD. It demonstrated the lack of association between AD and AMD, which is consistent with the findings of several recent clinical studies [10, 11]. Though AD and AMD seemingly share numerous features including an association with aging, common risk factors, and extracellular lesions containing Aβ and complement components, our results suggest the pathophysiology of these diseases is likely distinct. Aforementioned genetic studies further support this conclusion [1, 3, 33, 35, 36]. Though we did not observe any association between AD and AMD, there may be other ocular manifestations of the pathologic process occurring in AD. Identifying retinal biomarkers of AD appears promising and remains an area of active research [51, 52]. Potentially, due to recent advances in ocular imaging, the retina can provide a non-invasive means to earlier and more accurate diagnosis of AD.

## Acknowledgments

We would like to thank Brenda Dudzinski, who assisted with the creation of the autopsy database of cases; Christine Hulette, MD, who performed the neuropathologic diagnosis and AD

grading of brain specimens; and Alan Proia, MD, PhD, who performed the histopathologic evaluation and AMD grading of eye specimens used in this study.

## Author Contributions

**Conceptualization:** John R. Deans, P. Murali Doraiswamy, Heather E. Whitson, Eleonora M. Lad.

**Data curation:** Gordon J. Smilnak, Sandra Stinnett.

**Formal analysis:** Gordon J. Smilnak, Sandra Stinnett, Eleonora M. Lad.

**Funding acquisition:** Eleonora M. Lad.

**Investigation:** John R. Deans, P. Murali Doraiswamy.

**Methodology:** P. Murali Doraiswamy, Heather E. Whitson, Eleonora M. Lad.

**Project administration:** Eleonora M. Lad.

**Validation:** Sandra Stinnett.

**Writing – original draft:** John R. Deans, Heather E. Whitson, Eleonora M. Lad.

**Writing – review & editing:** Gordon J. Smilnak, P. Murali Doraiswamy, Eleonora M. Lad.

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
