## [Decision Letter · Decision Letter 0]

10 Jul 2019

PONE-D-19-17299

Comorbidity of age-related macular degeneration with Alzheimer’s disease: A histopathologic case-control study

PLOS ONE

Dear Dr. Lad,

Thank you for submitting your manuscript to PLOS ONE. After careful consideration, we feel that it has merit but does not fully meet PLOS ONE’s publication criteria as it currently stands. Therefore, we invite you to submit a revised version of the manuscript that addresses the points raised during the review process.

We would appreciate receiving your revised manuscript by Aug 24 2019 11:59PM. To enhance the reproducibility of your results, we recommend that if applicable you deposit your laboratory protocols in protocols.io, where a protocol can be assigned its own identifier (DOI) such that it can be cited independently in the future. For instructions see: http://journals.plos.org/plosone/s/submission-guidelines#loc-laboratory-protocols

We look forward to receiving your revised manuscript.

Kind regards,

Yi Su, Ph.D

Academic Editor

PLOS ONE

Journal Requirements:

2. We would be grateful if you could clarify whether the information included in the Supporting Information has been published previously, and if so, whether you have permission of the original copyright holder to reproduce these tables and images. If this information has been published previously we ask that you remove this information and include a relevant citation to the previous publications.

3. Please provide additional details regarding participant consent. Please state whether this was obtained from patients before death and/or from the next of kin. In the Methods section, please ensure that you have specified (1) whether consent was informed and (2) what type you obtained (for instance, written or verbal). If the need for consent was waived by the ethics committee, please include this information. Please also explain why the requirement for consent was waived.

4. We note that Figure 1 in your submission contain copyrighted images. All PLOS content is published under the Creative Commons Attribution License (CC BY 4.0), which means that the manuscript, images, and Supporting Information files will be freely available online, and any third party is permitted to access, download, copy, distribute, and use these materials in any way, even commercially, with proper attribution. For more information, see our copyright guidelines: http://journals.plos.org/plosone/s/licenses-and-copyright.

Additional Editor Comments:

This manuscript presented a retrospective pathology-based case-control study which investigates whether there is association between Alzheimer's disease (AD) and age-related macular degeneration (AMD). The study did not find a statistically significant association between AD and AMD and suggested that any shared characteristics between the two disease may be nondeterministic. This work received favorable review from expert reviewers and minor revision is recommended to address the issues raised by the reviewers.

Reviewers' comments:

Reviewer's Responses to Questions

**Comments to the Author**

1. Is the manuscript technically sound, and do the data support the conclusions?

Reviewer #1: Partly

Reviewer #2: Yes

2. Has the statistical analysis been performed appropriately and rigorously? 

Reviewer #1: I Don't Know

Reviewer #2: Yes

3. Have the authors made all data underlying the findings in their manuscript fully available?

Reviewer #1: Yes

Reviewer #2: Yes

4. Is the manuscript presented in an intelligible fashion and written in standard English?

Reviewer #1: Yes

Reviewer #2: Yes

5. Review Comments to the Author

Reviewer #1: This is study on a very important question in the field of neurodegeneration both for AMD and for Alzheimer.

The study is rather well done and the literature review and context placement is very well done. The authors note the limitations very well which are unavoidable in a study fo this nature that requires Brian and eye autopsy specimens. Thus despite the limitations this study adds significantly to our knowledge.

One eadditional control that may be added is that of a the AMD prevalence in a cohort of controls patients form the electronic data record with the same demographics as the AD autopsy group to cover for the imbalances noted by the authors in their control group. Of course this will have its own limitations but it will be nice t know this information if possible.

Thank you

Reviewer #2: This paper is a clearly written description of a methodologically rigorous study on an important topic. As the authors outline, the use of often elusive histopathological diagnoses makes this paper an impactful contribution to the field. It contains sufficient statistical information in general (though I wonder why the test statistic was given for chi-squared tests but not for t-tests), and thoughtful reflection in the discussion. I commend this write up as an excellent example of being as close to publication-ready as a submission can be. Well done!

- On page 15, line 310, change "Alike..." to "Like..." or similar.

- Some figures - if cost and space allow, which they may not - of examples of the histopathological findings would be engaging. Not essential.

- In the section, at the end of pages 14 and beginning of page 15, on the lack of shared polymorphisms between AMD and AD, consider citing this: Am J Geriatr Psychiatry. 2015 Dec;23(12):1290-1296. doi: 10.1016/j.jagp.2015.06.005 Its of direct relevance but my knowledge of this paper is because I am one of the authors, so clearly have an interest to declare on this point! The authors may well be aware of this paper and have decided against citing it, which is obviously fine. I also declare that I am the first author cited in reference 10.

6. PLOS authors have the option to publish the peer review history of their article (what does this mean?). If published, this will include your full peer review and any attached files.

Reviewer #1: Yes: Demetrios Vavvas

Reviewer #2: Yes: Dr Michael Williams

---

## [Author Response · Author response to Decision Letter 0]

20 Aug 2019

PONE-D-19-17299 

Comorbidity of age-related macular degeneration with Alzheimer’s disease: A histopathologic case-control study  We would like to thank the Academic Editor and the reviewers for taking the time to evaluate our manuscript and for their thoughtful, constructive comments. As requested, we have revised the manuscript. Below we detail our “point by point” responses to the reviews and we indicate the location of the modified material in the revised manuscript.  N/A   To enhance the reproducibility of your results, we recommend that if applicable you deposit your laboratory protocols in protocols.io, where a protocol can be assigned its own identifier (DOI) such that it can be cited independently in the future. For instructions see: http://journals.plos.org/plosone/s/submission-guidelines#loc-laboratory-protocols

N/A 

Please ensure that your manuscript meets PLOS ONE's style requirements, including those for file naming. The PLOS ONE style templates can be found at http://www.journals.plos.org/plosone/s/file?id=wjVg/PLOSOne_formatting_sample_main_body.pdf and http://www.journals.plos.org/plosone/s/file?id=ba62/PLOSOne_formatting_sample_title_authors_affiliations.pdf

2. We would be grateful if you could clarify whether the information included in the Supporting Information has been published previously, and if so, whether you have permission of the original copyright holder to reproduce these tables and images. If this information has been published previously we ask that you remove this information and include a relevant citation to the previous publications.

The tables included in the Supporting Information contain content that was previously published in the manuscripts referenced under 23-25. These references are cited appropriately in the Methods section (lines 115-120). The tables in the Supporting Information section were removed.

 3. Please provide additional details regarding participant consent. Please state whether this was obtained from patients before death and/or from the next of kin. In the Methods section, please ensure that you have specified (1) whether consent was informed and (2) what type you obtained (for instance, written or verbal). If the need for consent was waived by the ethics committee, please include this information. Please also explain why the requirement for consent was waived.

The need for participant consent was waived by the Duke IRB for this decedent research, as all research subject are deceased, and all personal health information (PHI) was used solely for research and was not disclosed to anyone outside the Duke University without removing all identifiers. 

This information was added to the Methods section (lines 135-139).

4. We note that Figure 1 in your submission contain copyrighted images. All PLOS content is published under the Creative Commons Attribution License (CC BY 4.0), which means that the manuscript, images, and Supporting Information files will be freely available online, and any third party is permitted to access, download, copy, distribute, and use these materials in any way, even commercially, with proper attribution. For more information, see our copyright guidelines: http://journals.plos.org/plosone/s/licenses-and-copyright.

We uploaded, as an “Other” file the rights license document granting permission to publish Figure 1 and the email confirmation from the copyright holder, SpringerNature. The copyright holder noted that, per legal department advice, SpringerNature is unable to sign any outside forms such as the Content Permission Form. However, the email communication states that we are able to reuse the figure in an open access publication under the Creative Commons Attribution License (CCAL) CC BY 4.0 with appropriate reference to the source in the figure legend.

This text was included under lines 151-2.

N/A

 N/A. The Supporting Information was removed, per # 2 above.   Additional Editor Comments:  This manuscript presented a retrospective pathology-based case-control study which investigates whether there is association between Alzheimer's disease (AD) and age-related macular degeneration (AMD). The study did not find a statistically significant association between AD and AMD and suggested that any shared characteristics between the two disease may be nondeterministic. This work received favorable review from expert reviewers and minor revision is recommended to address the issues raised by the reviewers.   Reviewers' comments:  Reviewer's Responses to Questions  Comments to the Author  1. Is the manuscript technically sound, and do the data support the conclusions?  The manuscript must describe a technically sound piece of scientific research with data that supports the conclusions. Experiments must have been conducted rigorously, with appropriate controls, replication, and sample sizes. The conclusions must be drawn appropriately based on the data presented.   Reviewer #1: Partly  Reviewer #2: Yes

 2. Has the statistical analysis been performed appropriately and rigorously?   Reviewer #1: I Don't Know  Reviewer #2: Yes

 3. Have the authors made all data underlying the findings in their manuscript fully available?  The PLOS Data policy requires authors to make all data underlying the findings described in their manuscript fully available without restriction, with rare exception (please refer to the Data Availability Statement in the manuscript PDF file). The data should be provided as part of the manuscript or its supporting information, or deposited to a public repository. For example, in addition to summary statistics, the data points behind means, medians and variance measures should be available. If there are restrictions on publicly sharing data—e.g. participant privacy or use of data from a third party—those must be specified.  Reviewer #1: Yes  Reviewer #2: Yes

 4. Is the manuscript presented in an intelligible fashion and written in standard English?  PLOS ONE does not copyedit accepted manuscripts, so the language in submitted articles must be clear, correct, and unambiguous. Any typographical or grammatical errors should be corrected at revision, so please note any specific errors here.  Reviewer #1: Yes  Reviewer #2: Yes

 5. Review Comments to the Author  Please use the space provided to explain your answers to the questions above. You may also include additional comments for the author, including concerns about dual publication, research ethics, or publication ethics. (Please upload your review as an attachment if it exceeds 20,000 characters)  Reviewer #1: This is study on a very important question in the field of neurodegeneration both for AMD and for Alzheimer. The study is rather well done and the literature review and context placement is very well done. The authors note the limitations very well which are unavoidable in a study for this nature that requires brain and eye autopsy specimens. Thus despite the limitations this study adds significantly to our knowledge. One additional control that may be added is that of a the AMD prevalence in a cohort of controls patients form the electronic data record with the same demographics as the AD autopsy group to cover for the imbalances noted by the authors in their control group. Of course this will have its own limitations but it will be nice to know this information if possible. Thank you

We would like to thank the reviewer for the very supportive comments. 

In order to attempt to fulfill this request, we completed a Duke Enterprise Data Unified Content Explorer (DEDUCE) search of the electronic data records from all normal, control patients seen at the Duke Medical Center over the past 37 years with the same demographics as the AD autopsy group (aged 75 and above). We found a number of 486,395 patients. Among these subjects, only 12,140 also carried the diagnosis of AMD, consistent with a prevalence of 2.49%, which is much lower and not consistent with estimates from prior studies (references 1, 27). This reflects the fact that only a minority of patients were seen at Duke Eye Center for a dilated examination that would allow a diagnosis of AMD to be made. A true prevalence of AMD in this population would only be possible if only the patients that received both a dilated eye examinations at Duke and a neurocognitive evaluation to evaluate AD were considered in this analysis. Unfortunately, these numbers are not available to us at this time. The reviewer’s astute suggestion was added to the Discussion section as an important future direction (lines 372-375):

“To control for the imbalances in demographics and comorbidities noted between the AD and non-AD group, a future direction would be the calculation of the AMD prevalence from the electronic data in a cohort of control patients record with the same demographics as the AD group.”

 Reviewer #2: This paper is a clearly written description of a methodologically rigorous study on an important topic. As the authors outline, the use of often elusive histopathological diagnoses makes this paper an impactful contribution to the field. It contains sufficient statistical information in general (though I wonder why the test statistic was given for chi-squared tests but not for t-tests), and thoughtful reflection in the discussion. 

We thank the reviewer for the positive and constructive feedback. 

The point on the test statistic is well taken, and for completeness, we are now reporting the test statistic for t-tests and z-tests in addition to chi-squared tests. 

I commend this write up as an excellent example of being as close to publication-ready as a submission can be. Well done! - On page 15, line 310, change "Alike..." to "Like..." or similar. 

 “Alike” was replaced with “Similar to” - Some figures - if cost and space allow, which they may not - of examples of the histopathological findings would be engaging. Not essential.

While a figure showcasing histopathological examples of stages of AMD and AD would be indeed engaging, this is not feasible due to the many panels necessary and current space limitations. However, we believe that the diagram in Figure 1 concisely illustrates the Sarks stages of AMD, and the definitions of Braak and Braak stages are included in the Methods section.

- In the section, at the end of pages 14 and beginning of page 15, on the lack of shared polymorphisms between AMD and AD, consider citing this: Am J Geriatr Psychiatry. 2015 Dec;23(12):1290-1296. doi: 10.1016/j.jagp.2015.06.005 Its of direct relevance but my knowledge of this paper is because I am one of the authors, so clearly have an interest to declare on this point! The authors may well be aware of this paper and have decided against citing it, which is obviously fine. I also declare that I am the first author cited in reference 10.

We appreciate this suggestion and have now cited this informative reference.

---

## [Decision Letter · Decision Letter 1]

17 Sep 2019

Comorbidity of age-related macular degeneration with Alzheimer’s disease: A histopathologic case-control study

PONE-D-19-17299R1

Dear Dr. Lad,

We are pleased to inform you that your manuscript has been judged scientifically suitable for publication and will be formally accepted for publication once it complies with all outstanding technical requirements.

With kind regards,

Yi Su, Ph.D

Academic Editor

PLOS ONE

Additional Editor Comments (optional):

Reviewers' comments:

Reviewer's Responses to Questions

**Comments to the Author**

1. If the authors have adequately addressed your comments raised in a previous round of review and you feel that this manuscript is now acceptable for publication, you may indicate that here to bypass the “Comments to the Author” section, enter your conflict of interest statement in the “Confidential to Editor” section, and submit your "Accept" recommendation.

Reviewer #1: All comments have been addressed

Reviewer #2: All comments have been addressed

2. Is the manuscript technically sound, and do the data support the conclusions?

Reviewer #1: Yes

Reviewer #2: Yes

3. Has the statistical analysis been performed appropriately and rigorously? 

Reviewer #1: Yes

Reviewer #2: Yes

4. Have the authors made all data underlying the findings in their manuscript fully available?

Reviewer #1: Yes

Reviewer #2: Yes

5. Is the manuscript presented in an intelligible fashion and written in standard English?

Reviewer #1: Yes

Reviewer #2: Yes

6. Review Comments to the Author

Reviewer #1: (No Response)

Reviewer #2: Thank you for addressing all the comments made, congratulations on this informative study and write up

7. PLOS authors have the option to publish the peer review history of their article (what does this mean?). If published, this will include your full peer review and any attached files.

Reviewer #1: Yes: Demetrios G Vavvas

Reviewer #2: Yes: Dr Michael Williams

---

## [Editor Report · Acceptance letter]

23 Sep 2019

PONE-D-19-17299R1 

Comorbidity of age-related macular degeneration with Alzheimer’s disease: A histopathologic case-control study 

Dear Dr. Lad:

I am pleased to inform you that your manuscript has been deemed suitable for publication in PLOS ONE. Congratulations! Your manuscript is now with our production department. 

With kind regards,

on behalf of

Dr. Yi Su 

Academic Editor

PLOS ONE